# Elementary School Teachers’ Perceptions of COVID-19-Related Restrictions on Food Allergy Management

**DOI:** 10.3390/nu14132714

**Published:** 2022-06-29

**Authors:** Mae Jhelene L. Santos, Natalie Riediger, Elissa M. Abrams, Nathalie Piquemal, Jennifer L. P. Protudjer

**Affiliations:** 1Department of Food and Human Nutritional Sciences, University of Manitoba, Winnipeg, MB R3T 2N2, Canada; santosmj@myumanitoba.ca (M.J.L.S.); Natalie.Riediger@umanitoba.ca (N.R.); 2Children’s Hospital Research Institute of Manitoba, Winnipeg, MB R3E 3P4, Canada; elissa.abrams@gmail.com; 3Department of Pediatrics and Child Health, Rady Faculty of Health Sciences, University of Manitoba, Winnipeg, MB R3A 1S1, Canada; 4Faculty of Education, University of Manitoba, Winnipeg, MB R3T 2N2, Canada; Nathalie.Piquemal@umanitoba.ca; 5George and Fay Yee Centre for Healthcare Innovation, Winnipeg, MB R3E 0T6, Canada; 6The Centre for Allergy Research, Karolinska Institute, 171 77 Stockholm, Sweden

**Keywords:** COVID-19, elementary school, food allergy, food allergy management, interviews, teachers, schools, qualitative study

## Abstract

(1) Background: Approximately 7% of Canadian children live with a food allergy (FA). Pre-COVID-19, ~20% of anaphylactic reactions occurred in schools. Yet, teachers reported poor FA-related knowledge, and experiences during the COVID-19 pandemic are not well-studied. Additionally, teachers’ management approaches vary widely. We aimed to describe elementary school teachers’ perceptions about FA management during the COVID-19 pandemic; (2) Methods: Using a semi-structured interview guide, English-speaking elementary school teachers in Winnipeg, Canada were interviewed virtually. Interviews were audio-recorded and transcribed verbatim. Data were analysed thematically; (3) Results: Most teachers were female and taught in public schools. Two themes were identified. Theme 1, COVID-19 restrictions made mealtimes more manageable, capturing the positive impacts of pandemic restrictions such as seating arrangements and enhanced cleaning. Limited lunchtime supervision prompted some teachers to assume this role. Theme 2, Food allergy management was indirectly adapted to fit changing COVID-19 restrictions, describing how changing restrictions influenced FA-related practices. FA training was offered virtually with less nursing support. Class cohorts and remote learning decreased teachers’ perceived risk and FA-related management responsibility; (4) Conclusions: COVID-19-related practices were perceived as positively influencing in-school FA management, although unintended consequences, such as increased supervisory roles for teachers and reduced nursing support, were described.

## 1. Introduction

Food allergy, defined by Boyce et al. (2010) as “a potentially life-threatening immunological response that occurs reproducibly upon ingestion of the allergen” [1], is a significant public health burden. Globally, food allergy affects an estimated 7–8% of children [2,3,4,5], corresponding to 1–2 students per average-sized Canadian classroom [6]. Food allergy triggers vary, as does reaction severity, which includes anaphylaxis, the most severe and potentially fatal allergic reaction [1].

Of all anaphylactic events amongst school-aged children, approximately 20% occurred in school settings, where children spend the majority of their waking hours [7,8]. A teacher’s primary role is to teach. Yet, teachers are also expected to care for children with health conditions, including but not limited to food allergies, albeit without adequate formal training [9]. Consequently, teachers have low and varied food-allergy-related knowledge and experience. Our group recently published a review on international, school-based food allergy management practices [10]. Therein, we identified the variability and heterogeneity in management practices amongst schools and the knowledge gaps amongst teachers, and teachers’ desires for more food allergy training and resources [10]. Due to limited resources and training available, it was unsurprising that teachers feel concern and anxiety related to managing chronic conditions in the school setting [9,10]. 

In Canada, at present, there is no universal outline or management plan available to guide schools even within jurisdictions under the same statute. In contrast to the Canadian provinces of Ontario and Alberta [11,12], no legislation surrounding the management of food allergy, including epinephrine auto-injector (EAI) administration, exists to protect children and teachers in the Province of Manitoba, the central Canadian province in which this study was conducted. Moreover, variation exists within schools in the same province. For example, at minimum, private institutions adopt provincial recommendations, but may choose to apply additional procedures. In contrast, some jurisdictions in the United States of America (USA) have legislation regarding EAI and the use of Emergency Anaphylaxis Plans (EAP) in schools. Yet, there is still a paucity and a general lack of standardized policies to inform the implementation of these [13]. In effect, policies are often at the discretion of school boards, the individual school, and the individual teacher. Frequently, policies include food restrictions [13]. However, these restrictions were not found sufficient to reduce the risk of accidental ingestion and allergic reactions [14,15], and checking every food item coming into the school setting was not feasible in practice [13]. 

In 2020, public health restrictions implemented to curb the spread of the coronavirus disease 2019 (COVID-19) undoubtedly changed the ways schools were governed. In Manitoba, among other jurisdictions, several public health restrictions were put forth in schools to reduce the spread of COVID-19. Such changes included the switch to remote learning, implementation of class cohorts, enhanced cleaning and subsequently, loosening of the restrictions. Reports by Mack et al. (2020) and Greenhawt et al. (2020) on COVID-19-related food allergy management practices highlighted the importance and need to explore the impacts of COVID-19 in school settings, specifically regarding food allergy management. Similarly, in October 2021, the Provincial Government of Manitoba released a document related to school re-opening and management for the school year 2021–2022 [16], which aligned with points outlined in previous reports [17,18]. Yet, little is known how these management strategies intended for schools impacted teachers, and specifically, if and how food allergy management was affected with these restrictions. Thus, the aim of this study was to describe elementary school teachers’ perceptions about food allergy management during the COVID-19 pandemic in Winnipeg, Canada.

## 2. Materials and Methods

This analysis was embedded within a larger study that aimed to describe the mental health impact and needs of children living with food allergy, as well as their caregivers, in Winnipeg, the capital city of the central Canadian province of Manitoba. Located 100 km from the Canada–USA border, Winnipeg is a city of ~760,000 people, representing half of Manitoba’s population [19]. Elementary schools in Winnipeg are public (government-funded without required fees) or private (funded in diverse ways, including tuition paid by families).

In the present analysis, we conducted qualitative interviews with teachers, who were recruited via social media and word-of-mouth between November 2021 and April 2022. During this data collection period, schools were open to in-person learning. Due to COVID-19-related restrictions, the return to in-person learning was delayed post-winter break, but schools re-opened by 10 January 2022 [20]. 

### 2.1. Eligibility Criteria 

Teachers must have been working in a Winnipeg-based public or private elementary (Kindergarten (K) to Grade 6) school, or who worked in a K-Grade 8 school and taught K-Grade 6. Teachers who were on leave (e.g., maternity leave), but held an employed teaching position, were also eligible. 

### 2.2. Interview Methods

Teachers were interviewed virtually by the project lead and student researcher (both initially blinded for review). Interviews were held via Microsoft Teams, in keeping with all COVID-19-related provincial and institutional public health guidelines. All participants were contacted via email and provided informed consent before participating.

We developed a semi-structured interview guide (see Appendix A) for this study. Teachers received a USD 30 e-gift card for their participation.

### 2.3. Theoretical Framework 

The study followed a pragmatic framework. Pragmatism was the chosen framework as it suits the objectives of this study, and the larger studies in which this present study was a part of, including identifying actionable ways to enhance food allergy management and the experiences of families living with food allergy. Pragmatism focuses on careful decision making that will meet the intended outcomes of the inquiry. Thus, the focus is on practical implications or applications, and allows the researcher to use methods, including data collection and analysis, that best solve real-world problems [21,22]. The researchers acknowledge that there are multiple realities based on socially constructed experiences and environments. Thus, the researchers’ worldview can influence the overall project. Overall, reality will be known using both objective and subjective evidence [21]. 

In practical use, pragmatism allowed the researchers to explore food allergy and to understand its unique meaning and significance by the teachers who were experiencing it, while the researchers attempted to interpret their experiences [21]. Through pragmatism, the researchers were able to identify key pieces of food allergy management amongst the different elementary grade levels, and subsequently, classify the actionable ways to have more available resources for teachers, as the researchers deemed appropriate and feasible in a real-life setting [22]. As the researchers were also active participants in the research process, careful consideration of data collection methods and various ways of reporting were deliberated [21,22]. 

### 2.4. Thematic Analysis 

Data were analyzed via thematic analysis, an active and inductive method to identify themes across a dataset [23]. This method aligned with our pragmatic approach, which offered flexibility and accessibility to identify units of data we deemed applicable to food allergy management from teachers’ perspectives [23]. Thematic analysis is a rigorous and organized method of analyzing summarizing data, which was merited in producing an in-depth yet approachable, narrative of patterns and meanings within the large and complex dataset [23]. 

Our analysis followed the six steps outlined in Braun and Clarke (2006)’s guide. The student researcher reviewed the transcripts multiple times and conducted two coding stages. The two researchers reviewed and subsequently collapsed the broad codes into themes. In keeping with the pragmatic framework in thematic analysis, data were analyzed inductively, then deductively, and simultaneously, during the coding process in order to gain a deep understanding of the data collected [23]. Thus, the researchers actively identified the themes within the data; themes were not emerging concepts [23,24]. Constructs were deemed saturated when new or additional constructs ceased to be identified with subsequent interviews. 

### 2.5. Rigor

Ongoing reflexivity and peer debriefing were carried out throughout the research process. Credibility was verified by thorough debriefing and data triangulation between the two researchers, having prolonged participant engagement, persistent observation during interviews, and member checking. Dependability and confirmability of the research was achieved through the use of data organization, field notes, and peer debriefing [25]. 

Transcripts were read twice in tandem with the audio recordings to ensure accuracy and increase research data familiarity. Verbatim quotes are included in the results to illustrate themes. Participants were de-identified and were instead reported as (T), followed by a meaningless identification code. This study was approved by the University of Manitoba Health Research Ethics Board (HS22242 (H2018:405)).

## 3. Results

We interviewed 16 teachers from various schools across Winnipeg (Table 1). Teachers were primarily female (14/16; 87.5%), worked in public schools (14/16; 87.5%), and taught earlier years students (K-Grade 3). Half of the teachers (8/16; 50.0%) taught in multiple and/or multi-grade classes. Half of the teachers (7/13; 53.8%) taught in lower-income areas, wherein >10.0% of households were living below the low-income cut-off, after tax [26,27]. Teachers had a mean teaching experience of 5.8 years (range 0.5–10 years) in private, public, and international institutions. Few teachers had supervisory roles (2/16; 12.5%). 

Interviews were, on average, 32 (range 20–45) minutes and were audio recorded and transcribed verbatim. Two themes were identified (Table 2).

### 3.1. Theme 1 “COVID-19 Restrictions Made Mealtimes More Manageable”

This theme captures how newly enforced precautions placed to decrease the spread of COVID-19 have indirectly and positively impacted food allergy management. Physically distanced seating arrangements, eating in the classroom, enhanced cleaning practices, and no outside food permitted enabled teachers to manage food allergy better, specifically at mealtimes. 

Teachers described relief in having “one less thing to worry about” (T8) as class parties were not allowed at the start of the pandemic. When class parties were allowed to resume, teachers asked parents to bring individually packaged foods to prevent food sharing. In schools with meal programs, wherein breakfast and/or lunch were available for students, only educational assistants (EA) or teachers served food as a pandemic-related measure to provide food safely to students. 

Enhanced cleaning initiatives were described as beneficial for food allergy management, including frequent cleaning using stronger disinfectants. One teacher reported being “a little more on it with kids washing their hands before and after eating” (T11). 

However, there were varying degrees of meal supervision available during the pandemic. Pre-pandemic, some schools had common eating areas where adults, often paid parents or EA, supervised students. During the pandemic, class cohorts and mandates eating in the classroom added mealtime supervising responsibilities onto some teachers. One teacher stated that, “If I leave the classroom, it’s chaos. So I choose myself to stay in the room during lunchtime” (T16), which was also attributable to the lack of lunchtime supervision available. During the pandemic, “there’s an EA (educational assistant) walking across the rooms.” (T16) Meanwhile, other schools, using the class cohort system, allowed fewer classes to eat in common spaces at the same time. Nevertheless, some teachers believed seating arrangements and eating in the classroom helped limit food sharing amongst students.

“It’s a lot easier for [teachers] to see, if [student] was handing a granola bar to a friend versus in the lunchroom, […] just makes a huge difference in managing behaviors that they’re [eating] in the classroom”.(T7)

### 3.2. Theme 2: “Food Allergy Management Was Indirectly Adapted to Fit Changing COVID-19 Restrictions”

Provincial school nurses provided virtual annual training, including, but not limited to, food allergy, which was “way less engaging to listen to compared to an actual in-person” training (T7). Teachers “were just sent the link to the video and instructed to watch it.” (T11). One teacher had trainer EAI provided by the school administrator to practice with. Teachers also reported less encounters and nursing support from the provincial program. “I have not personally seen the [provincial school nurse] this year. I don’t even recall seeing the [provincial school nurse] last year… maybe one time.” (T10).

Having class cohorts and remote learning decreased teachers’ perceived risk and management responsibility related to food allergy. 

“[Food allergy] is not something I considered specifically during COVID. Even though [COVID] kind of affected everything […] it might have gone to the back burner in a way because we didn’t have kids for so long. So when the kids are home, those spaces aren’t ours to worry about anymore. But then when the kids are back [in school], we’re thinking about sharing food more often”.(T20)

In Winnipeg, Manitoba, public health restrictions tightened and subsequently loosened during the data collection period [29]. As such, teachers also spoke of their experiences having to be flexible and adapt and change their practices as needed in order to reflect the provincial recommendations. As COVID-19-related restrictions were loosening, one teacher discussed upcoming changes regarding seating arrangements and physical barriers. 

“We’re gonna bring back tables. So 4–5 kids will sit per table and those shields will be gone […] so it’s gonna start to bring its challenges, I think, now as a result. […] In terms of food allergy […] those kids who have allergies will remain in desks, just because of space […] and to mitigate any contamination in that sense”.(T18)

## 4. Discussion

To our knowledge, this is the first study to explore elementary teachers’ perceptions of and experiences with food allergy management in the context of the COVID-19 pandemic. In this qualitative analysis, teachers spoke of the plurality of COVID-19 public health restrictions on food allergy management in Winnipeg elementary schools. Broadly speaking, teachers’ experiences managing food allergy changed during the COVID-19 pandemic. While positive perceptions about the pandemic-related practice were described, some restrictions unintendedly added an extra work burden. For example, teachers provided additional supervision at lunchtime, and spoke of the constant need for change and adaptation in practice as the COVID-19 pandemic-related restrictions went on. Teachers also agreed that enhanced cleaning protocols were beneficial for managing food allergy; however, they did not prefer the virtual training format.

A teacher’s primary role is to teach, yet teachers also play a key role in keeping children safe, regardless of whether children have a food allergy [9]. However, teachers may not have received adequate support and training during the pandemic, as a result of the redeployment of nursing staff due to COVID-19-related demands [30]. Pre-pandemic, teachers had poor baseline food allergy knowledge [10]. The lack of school nursing support and the virtual training model likely did not provide sufficient education for newer teachers, who also never experienced in-person training pre-pandemic. As this was unreported in our study, newer teachers likely had limited food-allergy-related experience and awareness, and/or comparison of pre-pandemic training. 

Moreover, teachers in our sample had varied experiences, including personal and indirect experience managing food allergy in their personal lives. This may have impacted teachers’ abilities to adapt their food allergy management practices to changing restrictions, as well as how teachers prioritized competing interests. One teacher described food allergy as being on the “back burner” during the pandemic, though thoughts about eating in school and managing allergies became more apparent as children transitioned back to in-person learning. In the literature, teachers described challenges related to emergency remote teaching transitions during the COVID-19 pandemic, including communicating with parents and students and engaging in learning [31]. During these transition periods, teachers may have prioritized delivering teaching materials to students virtually, in-person, and sometimes both. Teachers must manage many competing demands, that often are not food-allergy-related, which underscores the importance and need for food allergy emergency resources and support, as teachers may be unprepared in the case of an emergency. The inadvertent benefits of COVID-19-related restrictions also warrant consideration. Adapting the positive impacts of restrictions, such as enhanced cleaning protocols and no food sharing practices, may enhance food allergy management in schools and classrooms post-pandemic.

In this study, we purposively recruited Winnipeg-based elementary teachers. This type of recruitment allows for information-rich cases about the topic of interest, in this case, food allergy in schools [21]. Other teachers in our city, and indeed beyond, may have different experiences. Nevertheless, this study presents a diverse sample of Winnipeg-based teachers from various teaching backgrounds, school types, and jurisdictions. Despite the described heavy workloads of teachers during the pandemic, we were able to recruit 16 participants, which yielded a substantial amount of data, but also reflects teachers’ interest in food allergy management in schools. 

## 5. Conclusions

Teachers play a key role in food allergy management in school settings. As described, teachers had varied and changed experiences with managing food allergy during the COVID-19 pandemic. Though there were changes in resources and training available, there were nevertheless positive impacts of pandemic-related restrictions, such as enhanced cleaning and no food sharing, which ultimately helped teachers to manage food allergy in their classrooms. Post-pandemic, continuing these practices and providing more food-allergy-related training and resources will help teachers better manage food allergy in schools and classrooms.

## Figures and Tables

**Table 1 nutrients-14-02714-t001:** Participant and interview characteristics (N = 16).

		*n*	%
Sex	Female	14	87.5
	Male	2	12.5
School type	Public	14	87.5
	Private	2	12.5
Income level of school area *	Lower income	7	53.8
Grades taught **	Kindergarten-Grade 3	14	-
	Grade 4–6	5	-
Type of class	Single grade	8	50.0
	Multi-grade	8	50.0
Years of teaching experience	<5 years	5	31.3
	≥5 years	8	50.0
	Not reported	3	18.7
	Mean (Range; years)	13	5.8 (0.5–10)

* Participants (N = 13) taught in lower-income areas, wherein >10.0% of residents were reported living below the low-income cut-offs, after tax (LICO-AT) [28]. In this study, areas with >10% of residents living below the LICO-AT are considered lower-income areas, whereas areas with <10% of residents living below the LICO-AT are considered higher-income areas. Families who live below the low-income cut-offs after tax are those who are expected to spend more of their after-tax income on daily necessities, namely shelter, food, and clothing [27]. ** Does not total N = 16; some participants taught in multiple classes/grades.

**Table 2 nutrients-14-02714-t002:** Qualitative themes, summary statement, codes, and supporting quotations of COVID-19-related restrictions on food allergy management.

Theme 1: COVID-19 Restrictions Made Mealtimes More Manageable	
**Summary:** New rules such as seating arrangements, enhanced cleaning practices, and no outside food permitted enabled teachers to better manage food allergy, specifically at mealtimes. These precautions were placed to decrease the spread of COVID-19, but have indirectly positively influenced food allergy management.	
**Codes**	**Supporting Quotations**
Mealtime management	*“I think well, I don’t know if it’s just with allergies but I think that [pandemic] definitely helped in terms of what I usually did with micro-managing. So, because they were really on them [students] about not trading snacks or you know, we’re not sharing food, and we’re sitting further away, I feel like in terms of allergies… I’m at peace a little bit more, because I know that that’s what’s going on.”* (T2)
	*“I’ve had to resort to putting [children’s show] on YouTube for kids to watch so that they sit in their spots […] which has made me less worried about food allergies because I know that they’re not walking around […] and now they’re eating at their spots and watching. In that regards it has made [eating in the classroom] a little bit better, more manageable.”* (T16)
	*We don’t invite other foods in to give out for Halloween, or for birthdays or treats like that. I guess the difference now is that we don’t really have to manage that piece anymore when it comes to food too. I guess [treats are] one less thing to worry about.”* (T8)
Emphasis on cleanliness	*“I think the Accelerated hydrogen peroxide is more likely to remove nut oil residue from the surface than just soap and water.” *(T17)
	*“I’m a little more on it with kids washing their hands before and after eating, which is more so a COVID thing. But [hand washing] plays into the allergy as well.” *(T11)
**Theme 2: Food Allergy Management Was Indirectly Adapted to Fit Changing COVID-19 Restrictions**	
**Summary:** COVID-19-related restrictions influenced teachers’ food allergy management as school/classroom practices changed, such as the switch to virtual food allergy training, shift to remote learning, creation of class cohorts and subsequently, to loosening restrictions. Teachers had to be flexible and adapt to restrictions as they changed.	
**Codes**	**Supporting Quotations**
Modified URIS training and resource provision	*“I have not personally seen the [provincial school nurse] this year. I don’t even recall seeing the [provincial school nurse] last year… maybe one time.”* (T10)
	*“[Administrator] just had the [auto-injector], and it’s like if you want to practice injecting cause on the video they did show us how to do it. We just didn’t have the actual physical thing. Normally [provincial school nurse] come in show us how to do it and we all have to do it kinda thing […] just have been the principal observing us […] I don’t even know if she would’ve been there, or if it was just in her office.”* (T7)
Physical division	*“Kids who do have those particular allergies will remain in desks, just space-wise and to mitigate any contamination in that sense. We are keeping the ones who don’t have allergies on tables and the ones who do [have allergies] on desks around the tables and just kinda spacing them out.”* (T18)
	*“Because of cohorts, and students can’t drift between cohorts so easily, so someone else being affected by someone’s food because of food allergy is way lower, in my opinion.”* (T13)
Changing COVID restrictions	*[Class party] was a thing before COVID […] last year, we didn’t allow kids to bring in any birthday treats or anything. This year we’ve been okay if they’re individually packaged and the box is unopened from the store.* (T11)
	*I mean, [food allergy] is not something I considered specifically during COVID. Even though it’s kind of affected everything […] it might have gone to the back burner in a way because we didn’t have kids for so long. So when the kids are home, those spaces aren’t ours to worry about anymore. But then when the kids are back [in school], we’re thinking about sharing food more often.”* (T20)

## Data Availability

Owing to the senstive nature of the qualitative data, and the potential for participants to be recognised, data will not be made available.

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
