# Peer review of "Elementary School Teachers’ Perceptions of COVID-19-Related Restrictions on Food Allergy Management"

_nutrients, 2022, doi:10.3390/nu14132714_

Round 1

Reviewer 1 Report

please state better the choice of pragmatic as a theorethical framework, especially state better why this theoretical framework is in your opinion accurate (108-117)

same for the choice of thematic analysis and especially its application to pragmatism (119-125) 

Author Response

Comment 1

please state better the choice of pragmatic as a theorethical framework, especially state better why this theoretical framework is in your opinion accurate (108-117)

Response to Comment 1

Thank you for taking the time to review our paper and for the comments. Changes related to the comment were made on the manuscript as attached, please see lines 118-137.

Our decision to follow a pragmatic framework was largely driven by the desire to answer the main objectives of the overarching studies in which this project is part of. The aim of the primary CIHR-funded study was to describe the mental health impact and needs of children living with food allergy, as well as their caregivers, in Winnipeg, Canada. As part of this study, caregivers/ parents, healthcare providers and teachers were all interviewed to gain different perspectives to inform this study. As such, this present study, as part of Mae Santos’ thesis project focused on teachers’ experiences and perceptions of food allergy management in schools, where children spend most of their waking hours.

Given the patient-oriented and practical focus of the overarching study and the larger thesis project, the framework of pragmatism lends value, and in our opinion, suitable, for this type of inquiry, as pragmatism is informed and is focused on “what works” to find and choose the best methods for the research process (Creswell, 2013). As is the nature of naturalistic inquiry, “being open and pragmatic requires a high tolerance for ambiguity and uncertainty as well as trust in the ultimate value of what inductive analysis will yield.” (Patton, 2002)  In addition, more recent publications on the utility of pragmatism in mixed methods, and patient-oriented research has also been described as a framework that can bring voice to marginalized communities, which have further illuminated and solidified our choice in using a pragmatic framework for this study (Allemang et al., 2021; Kaushik & Walsh, 2019; Ramanadhan et al., 2021)

Comment 2

same for the choice of thematic analysis and especially its application to pragmatism 139-158) 

Response to Comment 2

Thank you for this inquiry. Changes related to the comment were made on the manuscript as attached, please see lines 138-157.

In our experience carrying out this present study, utilizing a pragmatic framework and thematic analysis, guided by Braun & Clarke (2006), was advantageous, especially in the midst of changing COVID-related environments. As the objectives of the thesis project, as well as the overarching study in which this present study was a part of, teachers were asked about their experiences before and during the COVID-19 pandemic. There were many “moving parts” to the management of food allergy in their classrooms and schools, during these different times (i.e., pre-, during and post-pandemic), an in-depth and thorough understanding and analysis was necessary to produce a substantial qualitative report.

Using thematic analysis allowed for exploration of many avenues of inquiry related to teachers’ food allergy management in schools during these time periods, due to the flexible nature of thematic analysis. Specifically, thematic analysis does not prescribe one specific data collection method, theoretical position and/or frameworks yet is still able to answer various types of qualitative inquiries (Braun & Clarke, 2013). Lastly, thematic analysis is accessible, both to new and seasoned researchers, in various disciplines and practices (Braun & Clarke, 2006). As such, data reporting is organized, concise and allowed the researchers to incorporate sound and appropriate research decisions in the process while adhering to practices that maintain rigor of the research (and physically distanced measures following public health guidelines). Once again, similar characteristics of flexibility and choosing what is deemed appropriate by the researchers to answer the research question and fulfill the aims of the study were factors that helped the researchers choose a pragmatic framework and thematic analysis for this study.

Reviewer 2 Report

In this manuscript, the authors conducted a survey on the elementary school teachers' opinions on food allergy management in the background of COVID19 pandemic. The background was clear and the survey methods were appropriate. There are a few grammar mistakes which need to be revised before publication.

Author Response

Comment 1

In this manuscript, the authors conducted a survey on the elementary school teachers' opinions on food allergy management in the background of COVID19 pandemic. The background was clear and the survey methods were appropriate. There are a few grammar mistakes which need to be revised before publication.

Response to Comment 1

Thank you for your time and effort in reviewing our paper. Lines 51, 54,  123-128, 175, 179, 189, 190-191, 206, 208, 235-236, 305, 318 have been revised in the attached manuscript.

References

Allemang, B., Sitter, K., & Dimitropoulos, G. (2021). Pragmatism as a paradigm for patient‐oriented research. 25, 38–47. https://doi.org/10.1111/hex.13384

Braun, V., & Clarke, V. (2006). Using thematic analysis in psychology. Qualitative Research in Psychology, 3(2), 77–101. https://doi.org/10.1191/1478088706qp063oa

Braun, V., & Clarke, V. (2013). Successful qualitative research: A practical guide for beginners. Sage Publications, Inc.

Creswell, J. W. (2013). Philosophical assumptions and interpretive frameworks. In Qualitative inquiry and research design: Choosing among five approaches (2nd ed., pp. 15–41). Sage Publications, Inc.

Kaushik, V., & Walsh, C. A. (2019). Pragmatism as a Research Paradigm and Its Implications for Social Work Research. Social Sciences, 8(9), 255. https://doi.org/10.3390/socsci8090255

Patton, M. Q. (2002). Qualitative Research & Evaluation Methods (3rd ed.). Sage Publications, Inc.

Ramanadhan, S., Revette, A. C., Lee, R. M., & Aveling, E. L. (2021). Pragmatic approaches to analyzing qualitative data for implementation science: An introduction. Implementation Science Communications, 2(1), 70. https://doi.org/10.1186/s43058-021-00174-1